# Recent Progress in Development of Carbon-Nanotube-Based Photo-Thermoelectric Sensors and Their Applications in Ubiquitous Non-Destructive Inspections

**DOI:** 10.3390/mi14010061

**Published:** 2022-12-26

**Authors:** Kou Li, Yuya Kinoshita, Daiki Sakai, Yukio Kawano

**Affiliations:** 1Laboratory for Future Interdisciplinary Research of Science and Technology, Tokyo Institute of Technology, 2-12-1 Ookayama, Meguro-ku, Tokyo 152-8552, Japan; 2Department of Electrical and Electronic Engineering, Tokyo Institute of Technology, School of Engineering, 2-12-1 Ookayama, Meguro-ku, Tokyo 152-8552, Japan; 3Department of Electrical, Electronic, and Communication Engineering, Faculty of Science and Engineering, Chuo University, 1-13-27 Kasuga, Bunkyo-ku, Tokyo 112-8551, Japan; 4National Institute of Informatics, 2-1-2 Hitotsubashi, Chiyoda-ku, Tokyo 101-8430, Japan

**Keywords:** carbon nanotubes, photo-thermoelectric effect, broadband photo sensing, flexible and stretchable electronics, non-destructive imaging inspection

## Abstract

The photo-thermoelectric (PTE) effect in electronic materials effectively combines photo-absorption-induced local heating and associated thermoelectric conversion for uncooled and broadband photo-detection. In particular, this work comprehensively summarizes the operating mechanism of carbon nanotube (CNT)-film-based PTE sensors and ubiquitous non-destructive inspections realized by exploiting the material properties of CNT films. Formation of heterogeneous material junctions across the CNT-film-based PTE sensors, namely photo-detection interfaces, triggers the Seebeck effect with photo-absorption-induced local heating. Typical photo-detection interfaces include a channel–electrode boundary and a junction between P-type CNTs and N-type CNTs (PN junctions). While the original CNT film channel exhibits positive Seebeck coefficient values, the material selections of the counterpart freely govern the intensity and polarity of the PTE response signals. Based on these operating mechanisms, CNT film PTE sensors demonstrate a variety of physical and chemical non-destructive inspections. The device aggregates broad multi-spectral optical information regarding the targets and reconstructs their inner composite or layered structures. Arbitrary deformations of the device are attributed to the macroscopic flexibility of the CNT films to further monitor targets from omni-directional viewing angles without blind spots. Detection of blackbody radiation from targets using the device also visualizes their behaviors and associated changes.

## 1. Introduction

Photo-sensing or photo-imaging measurements allow non-contact acquisition of optical information for monitoring targets in a large area. The optical properties of various objects are typically variable in monitoring wavelengths or bands. This inherent nature of photo-irradiation enables hyper-spectral material identification of composite objects by aggregating optical properties at multiple wavelengths [1,2] or non-contact structural reconstructions of layered targets by analyzing the degree of transmission, reflection, and absorption against illumination [3,4,5]. Therefore, photo-sensing or photo-imaging measurements advantageously play a leading role in techniques for non-destructive inspections among other types of testing methods [6,7,8,9]. To handle diverse materials or objects consisting of these in non-destructive inspections, effective use of longer-wavelength light over typical X-ray, ultraviolet (UV), and visible light (Vis) bands is a potential approach. Here, longer-wavelength light includes millimeter- or terahertz-waves (MMW, THz) and infrared light (IR: far-IR (FIR), mid-IR (MIR), and near-IR (NIR)). Irradiation in these bands exhibits transparency to even non-metallic opaque objects compared to Vis, with much lower invasiveness than with X-rays. Fingerprint spectra or intrinsic absorption characteristics often exist in these bands (e.g., polymers and gases) [10,11], and such features also enrich non-destructive inspection techniques.

To perform fundamental measurements in the above electromagnetic-wave regions, careful selection of the photo-detection mechanism is indispensable. Typical MMW–IR detection methods include electronic-, photon-, and thermal-type approaches. Schottky barrier diodes, CMOS devices, and high-electron-mobility transistors are representative examples of electronic-type detectors [12,13,14]. Electronic-type devices function with an ultrafast photo-detection speed (order of a few picoseconds) and high sensitivities, particularly in sub-THz regions in uncooled conditions. Photon-type devices, such as quantum well and quantum dot structures, function with a fast photo-detection speed (order of several tens of picoseconds) and ultrahigh sensitivities at extremely low operating temperatures [15,16]. Bolometers and Golay cells typically serve as thermal-type devices and perform uncooled ultrabroadband photo-detection ranging from the sub-THz to IR regions [17]. Under the motivation to deal with a wider range of inspection materials and objects, employing the thermal-type photo-detection mechanism is effective because of its advantageous characteristics: ultrabroadband measurements over the electronic-type and uncooled operations over the photon-type. First, ultrabroadband photo-detection is essential for aggregating multi-spectral optical information of inspection targets. Uncooled operations facilitate compact and portable systematization without the need for bulky cooling equipment. This feature is indispensable for expanding the range of evaluable targets by relaxing their inherent location and environmental limitations. Thermal-type photo-detection includes bolometric detection and the photo-thermoelectric effect (PTE effect). Device operations under the PTE effect function without applying a voltage bias and allow photo-detection with a lower limit of thermal noise. By contrast, bolometric detection requires an external voltage bias and induces 1/f noise along with thermal noise [18]. Therefore, the device design and the associated system development of PTE effect-based photo-sensing or photo-imaging techniques potentially enrich the existing non-destructive inspection platforms.

The PTE effect generally synergizes two different energy conversion phenomena: photo-thermal and thermoelectric. The PTE effect induces thermoelectric conversion at the photo-detection interface with local heating owing to photo-absorption. This means that utilizing the PTE sensor detects external irradiation by obtaining electrical response signals. When the PTE sensor consists of a single material structure, the photo-detection interface locates at the edge of the channel to localize the photo-induced heating. Here, the PTE response is proportional to the Seebeck coefficient of the device material and photo-induced temperature gradient across the channel [19]. On the other hand, the PTE sensor consisting of multiple materials often exhibits its photo-detection interface in the bonding junction. This is because a difference between each material’s Seebeck coefficient corresponds to the effective Seebeck coefficient of the bonding junction, and proper material combinations lead to enhancing the PTE response [20]. Therefore, the PTE device design includes not only device material evaluations from the viewpoint of photonic, thermal, and electronic aspects but also material combinations and their structural management, allowing a wide range of investigation approaches. Indeed, today’s investigation field of the PTE device design is rapidly growing [21,22,23], and various materials exhibit their potential suitability for PTE conversion, as described later. However, development of PTE sensors satisfying sensitive photo-detection in the aforementioned MMW–NIR bands and the associated demonstrations of ultrabroadband multi-wavelength non-destructive inspections are still insufficient.

Based on these considerations, this review introduces a series of contributions of PTE device design strategies with carbon nanotube (CNT) films to demonstrate multi-functional non-destructive inspections (Figure 1). Comprehensive assessments of the CNT-film-based PTE conversion aim to maximize the material properties for satisfying sensitive ultrabroadband photo-detection and further effectively apply the inherent structural nature: thin-film with mechanical softness for omni-directional monitoring of 3D objects without blind spots. The following chapters break down each device design strategy from describing the theoretical background to clarifying the respective contributions from material, structural, mechanical, and optical approaches to multi-functional performances of the CNT film PTE sensors or ubiquitous non-destructive inspection platforms.

## 2. Sensor Material: CNT Films

As the first step in designing PTE sensors, this chapter breaks down the fundamental properties of CNT films as the device material (Figure 2a–c). CNT films consist of randomly stacked CNTs and their network structures, resulting in high mechanical strength [27] and flexibility [28]. The CNT film also effectively absorbs ultrabroadband photo-irradiation at room temperature [29]. These features emphasize that employing a CNT film plays an essential role in developing ubiquitous non-destructive inspection devices and systems among other representative materials for PTE conversion, such as graphene [30], CNTs [31], MoS_2_ [32], SnSe [33], NbS_3_ [34], and PEDOT:PSS [35]. The CNT film collectively and advantageously satisfies the macroscopic free deformations and ultrabroadband photo-absorption among the above candidates. The CNT film further exhibits the Seebeck coefficients ranging from tens to hundreds of µV/K, which are comparable to those of PEDOT:PSS and so on. Based on these, utilizing the CNT film effectively converts external ultrabroadband photo-irradiation power to electrical signals for sensing or imaging measurements and simultaneously provides omni-directional viewing angles to 3D-structure targets.

Figure 2d presents the current-voltage characteristics of the CNT-film-based PTE sensor with and without external light irradiation to the device. The result clearly shows ohmic contacts between the CNT film channel and metal electrodes for signal readout, and this behavior allows PTE conversion under zero-bias-voltage operations, as mentioned earlier. As detailed in the following chapters, this review focuses on single-walled semiconducting-metallic-mixed CNT films. The overlapping of each energy band structure of randomly stacked CNTs is a possible factor for the ohmic contacts of the single-walled semiconducting-metallic-mixed CNT films with diverse metal electrode materials.

Finally Figure 2e,f introduces the fundamental patterning techniques of the CNT film. As dispersion solutions of CNTs are widely available, suction filtration is a possible approach. Employing laser-processed thin-film masks allows selective CNT film patterning (Figure 2e). This method freely controls the shape, size, and positions of the CNT films via laser processing on the mask. Management of the mask’s thickness and the solution’s drop amount also leads to thickness control of the CNT film up to micrometer-order. While efficient photo-absorption requires micrometer-order thickness of CNT films [36], the above thickness control eases the device design. Other representative CNT film patterning approaches include lithography (nanometer-order thickness) [37] and manual cutting (up to millimeter-order thickness) [38]. Although manual cutting potentially contributes to fabricating the PTE sensors for its available thickness range, the presented self-aligned suction filtration process advantageously realizes higher-yield repeatable micrometer-order spacing or patterning of the CNT film over manual handling. On the other hand, management of the solution’s drop amount controls the friction force between the mask and a membrane filter (Figure 2f). This means that the self-aligned suction filtration of the CNT film enables both direct patterning on the membrane and embedding in the mask film. Therefore, the presented approach freely develops CNT film patterning on various thin-film materials regardless of their water absorbability.

## 3. PTE Effect in Channel–Electrode Junctions

Based on the above device material considerations, this and the following chapter summarize the representative CNT-film-based PTE conversion: mechanism modeling, design strategies, and fundamental performances [24,25,26]. As thermoelectric conversion often employs the Seebeck effect at heterogeneous material junctions (typically P-type and N-type materials) to enhance the effective Seebeck coefficients, there are mainly two types of device structures for the CNT film PTE sensor. One is a CNT film channel–metal electrode boundary structure [24] and the other is a PN-junction-channel structure [25]. This chapter introduces the former type.

### 3.1. Structural and Material Approach

As CNT films exhibit positive Seebeck coefficients in the original state [38], employing metal electrode materials with negative Seebeck coefficients (e.g., bismuth (Bi): −70 µV/K and nickel: −20 µV/K) is effective [39]. Figure 3a shows that structural approaches are essential for governing the effective Seebeck coefficients of the device, prior to such material approaches. There are readout metal electrodes on the CNT film, typically in a direction parallel to the photo-induced thermal gradient across the channel: “Parallel” in the figure [40]. In this case, the following equation describes the associated PTE effect [24]:(1)$$\Delta V=\left|S_{\mathrm{CNT}}-S_{\mathrm{Com}}\right|\times \Delta T$$
here, Δ*V*, *S*_CNT_, *S*_Com_, and Δ*T* correspond to the DC-voltage PTE response, Seebeck coefficients of the CNT film and composite body between the channel and electrode, and photo-induced thermal gradient across the channel, respectively. The next equation further breaks down *S*_Com_ as follows [24]:(2)$$S_{\mathrm{Com}}=\frac{\sigma_{\mathrm{CNT}}t_{\mathrm{CNT}}S_{\mathrm{CNT}}+\sigma_{\mathrm{Metal}}t_{\mathrm{Metal}}S_{\mathrm{Metal}}}{\sigma_{\mathrm{CNT}}S_{\mathrm{CNT}}+\sigma_{\mathrm{Metal}}S_{\mathrm{Metal}}}$$
where *σ* and *t* correspond to electrical conductivity and thickness, respectively. As the thicknesses of the channel and electrode significantly affect *S*_Com_, *t*_CNT_ and *t*_Metal_ typically range from a few to several tens of micrometers and several tens of nanometers. This situation leads to an approximation of *S*_Com_ to *S*_CNT_ and causes attenuation of the effective Seebeck coefficient and associated PTE response of the device. Meanwhile, the series coupling of the CNT film channel and metal electrode fully utilizes their fundamental Seebeck coefficient values, enhancing the effective Seebeck coefficient of the device. Figure 3b shows the photo-responses of the device equipped with both series and parallel channel–electrode coupling structures. The PTE conversion in the series coupling interface enhances the device’s photo-response by five times that obtained in the parallel coupling interface. These behaviors are in good agreement with measured Seebeck coefficient values of each device component: *S*_Eff-Series_ 60.4 µV/K (*S*_CNT_ 62 µV/K − *S*_Au_ 1.6 µV/K) and *S*_Eff-Parallel_ 12.1 µV/K (*S*_CNT_ 62 µV/K − *S*_Com_ 49.9 µV/K) [24]. The above structural approach then contributes to the material approach in designing the CNT-film-based PTE sensors, as shown in Figure 3c. In the series coupling structure, proper selection of the metal electrode material regarding its Seebeck coefficient maximizes the effective Seebeck coefficient of the device. Indeed, employing the series coupling of the CNT film Bi electrode further enhances the device PTE response by two times that obtained in the channel-metal (positive Seebeck coefficient value, e.g., Au) series coupling [24]. Figure 3d,e shows the Au thin film lamination (20 nm thick) over the series electrode (200 nm thick) to relax Bi’s inherent significant oxidation in air [41]. The Au thin-film lamination serves as both adhesion to the CNT film channel and oxidation protection. The device with the Au-Bi-laminated series electrode structure exhibits lower electrical resistance value and time-dependent stability in air compared with those of the device with the pristine series Bi electrode (Figure 3e) [24]. Reductions in the electrical resistance of the device greatly contribute to managing the photo-detection sensitivity, namely the noise equivalent power (NEP) [38]:(3)$$\mathrm{NEP}=\frac{V_{\mathrm{Noise}}}{V_{\mathrm{Sensitivity}}}=\frac{V_{\mathrm{Thermal}\;\mathrm{noise}}}{V_{\mathrm{Sensitivity}}}=\frac{\sqrt{4k_{B}TR}}{S_{\mathrm{Eff}}\times \Delta T}\times P_{\mathrm{Eff}}$$
where *k*_B_, *T*, *R*, and *P*_Eff_ are the Boltzmann constant, absolute temperature, electrical resistance, and effective output power of light irradiation on the photo-detection interface, respectively.

### 3.2. Size Approach

In addition to the above structural and material approaches, the typical PTE sensor design strategy includes device size optimization. Because device size governs the electrical and thermal resistances, size optimization is essential to suppress thermal noise and maintain or maximize the photo-induced temperature gradient and the associated PTE response at the photo-detection interface. This section focuses on optimization of electrode length and device width. Figure 3f,g shows that the optimal series electrode length matches well with the photo-induced heat diffusion length in the device [24]. When the series electrode length is longer than the photo-induced heat diffusion length, there are excessive increments in the thermal noise (Figure 3g: from electrical resistance) and degradation in the NEP value of the device. Therefore, shortening the series electrode length is indispensable for gradually reducing the corresponding electrical resistance and thermal noise. On the other hand, shortening the series electrode length simultaneously reduced the thermal resistance at the photo-detection interface. Therefore, an insufficient photo-induced temperature gradient degrades the associated device PTE responses and NEP values when the series electrode length is shorter than the heat-diffusion length. The changes in the PTE responses and NEP values of the device are in good accordance with the numerical estimations (Figure 3f), and the optimal series electrode length (3.5 mm: with the smallest NEP value) is of the same order of magnitude as the heat diffusion length in Bi (1.3 mm) [42,43,44].

Together with length optimization, device width evaluation is essential to govern the PTE sensor performance [40]. Figure 3h–j summarizes the width optimization of the CNT-film-based PTE sensor with the series electrode structure. The following equation first breaks down the PTE response regarding the contributions of the device width to the photo-detection sensitivity [24]:(4)$$\Delta V=S_{\mathrm{Eff}}\times \Delta T\propto R_{\mathrm{Thermal}}\propto \frac{1}{w}$$
where *R*_Thermal_ and *w* correspond to the thermal resistance and device width, respectively. Based on the above, the following equation further clarifies the relationship between device width and NEP values [24]:(5)$$\mathrm{NEP}\propto \frac{\sqrt{R_{\mathrm{Electrical}}}}{S_{\mathrm{Eff}}\times \Delta T}\propto \frac{\frac{1}{\sqrt{w}}}{\frac{1}{w}}=\sqrt{w}$$
where *R*_Electrical_ corresponds to the electrical resistance. These situations imply that designing a device with finer width results in improvements in both the PTE responses and NEP values (Figure 3i,j). Here, the optimal device width is approximately 250 µm with respect to the wavelength range in the MMW–NIR regions [24]. In this size approach in designing the CNT-film-based PTE sensor, numerical estimations employ steady-state thermal distribution simulation by solving the following heat conduction equation via the finite element method (FEM) [40]:(6)$$\rho C\frac{\partial T}{\partial t}=k\left(\frac{\partial^{2}}{\partial x^{2}}+\frac{\partial^{2}}{\partial y^{2}}+\frac{\partial^{2}}{\partial z^{2}}\right)T+Q$$
where *ρ*, *C*, *k*, and *Q* are density, heat capacity, thermal conductivity, and total heat flow, respectively.

### 3.3. Performances

By incorporating the above structural, material, and size approaches, the device exhibits the minimum NEP value of 5.06 pWHz^−1/2^ as the photo-detection sensitivity [24]. This performance is 110 times the photo-detection sensitivity of the pristine CNT-film-based PTE sensor: parallel channel–electrode coupling structure with a device width of 1 mm (Figure 3k: transmissive XY scanning of a concealed metallic clip under FIR irradiation with a single-pixel device). The obtained photo-detection sensitivity is also comparable with that of cutting-edge solid-state photo-detectors, such as the bolometer [45,46] and Schottky barrier diodes [47]. The obtained 110-times photo-detection sensitivity enhancement involves six factors: series channel–electrode coupling, employing Bi, electrode length optimization, conformal coating of the series electrode, and device width optimization. NEP values of the device at each step are 560, 110, 53, 27.2, 17, and 5.06 pWHz^−1/2^, providing contributions of 34%, 14%, 7%, 23%, and 22%, respectively [24]. This means that the series PTE coupling between the CNT film channel and metal electrode with the higher Seebeck coefficient of negative polarity dominantly affects the device design strategies regarding enhancement of photo-detection sensitivity. Therefore, enhancing the Seebeck coefficient of the CNT film channel and employing thermoelectric materials of Bi compounds as the series electrode [48] potentially realizes further photo-detection sensitivity improvements. Here, the sensitivity limit of the device is approximately 45 fWHz^−1/2^ by designing superior series PTE coupling: carrier-doped semiconducting CNT films and Bi_2_Te_3_ (from ZT_CNT_: 0.03 to ZT_Semi-CNT_: 0.33, from ZT_Bi_: 0.1 to ZT_Bi2Te3_: 1) [49,50,51]. Compared with the CNT-film-based PTE sensor with the PN junction (in Chapter 4), the presented structure advantageously allows device operations with a higher spatial resolution (Figure 3l). Although the PN-junction-type PTE sensor consists of fully soft materials and even enables stretchable device configurations, the photo-detection interface within the CNT film channel hinders localization of photo-induced heating. Meanwhile, the presented coupling structure confines external irradiation to the device via surface reflection with the series electrode and the parallel electrode on the opposite side. This leads to heat localization at the photo-detection interface, and the associated PTE response mapping of the device is 8.7 times finer than that of the PN-junction type [24]. This is an example of the functionality difference in the CNT-film-based PTE sensors with each device design strategy, and it is inferred that the proper choice of device structure is available depending on the use cases.

## 4. PTE Effect in PN Junctions

The above chapter proved that the PTE design of the CNT-film-based photo-detector allows device operation with sensitivity comparable to that of solid-state photo-detectors. This chapter then introduces the fundamentals of the CNT-film-based PTE sensor with the PN-junction structure [25]. As this type of device consists of fully soft materials, a deeper understanding of mechanism modeling, optimization strategies, and performance in the following sections leads to realization of shape-deformable and shape-conformable photo-monitoring applications.

### 4.1. Mechanism

Figure 4a–c describes the fundamental operating mechanism of the CNT-film-based PTE sensor with the PN-junction structure. This device structure employs chemical liquid coating with a mixture solution of hydroxide and crown ether [52] for simple handling and atmospheric stability at room temperature among numerous N-type charrier doping methods of CNT films [53,54,55]. The process requires dopant application onto the half-side of the channel (Figure 4b), and the corresponding region immediately exhibits the Seebeck coefficient of the negative polarity after a few tens of seconds of drying (Figure 4c). Here, the dopant serves as follows: crown ether and anions form complexes with each other, and freely functional cations (OH^−^) inject electrons into the CNT film (Figure 4a). The previous work clarifies that the presented chemical carrier doping functions stably for at least one month [56]. In this structure, the effective Seebeck coefficient in the photo-detection interface equals a difference in Seebeck coefficients of the P-type and N-type CNT film channels. In other words, forming the PN junction as the photo-detection interface maximizes the effective Seebeck coefficients by summing up the absolute values of the Seebeck coefficients of the P-type and N-type CNT film channels [25].

### 4.2. Material Approach

Because the PN junction structure consists of the CNT film itself, material approaches on the channel play an essential role in this device design. In particular, the electronic state of CNTs, which varies with chirality and structure, has a significant impact on the performance of PTE sensors. Figure 4d–g shows the material properties for different CNTs types and the associated device performances, clarifying that use of single-walled semiconducting-metallic-mixed CNTs maximizes the photo-detection sensitivity [25]. Under the operating mechanism of the PTE effect, the device potentially has three candidate CNTs types: single-walled semiconducting (for its efficient thermoelectric conversion), multi-walled (for its efficient photo-absorption with easier fabrication and relatively inexpensive cost over single-walled types), and single-walled semiconducting-metallic-mixed. Despite the semiconducting CNT film having the highest Seebeck coefficient (Figure 4d), its lowest photo-absorption hinders the photo-detection of the device (Figure 4f) [25]. The thermal noise of the semiconducting CNT film (Figure 4e), associated with lower electrical conductivity, also degrades the photo-detection sensitivity (Figure 4g). Although the multi-walled CNT film exhibits the highest photo-absorption characteristics (Figure 4f), its lowest Seebeck coefficient (Figure 4d) attenuates the photo-response of the device (Figure 4g) [25]. On the other hand, the device still functions with a minimum NEP value comparable to the aforementioned solid-state photo-detectors by employing a single-walled semiconducting-metallic-mixed CNT film channel. The NEP values of the CNT-film-based PTE sensor in broad frequency ranges are as follows: 236 pWHz^−1/2^ in MMW band (*λ* = 1.15 mm), 105 pWHz^−1/2^ in sub-THz band (*λ* = 577 μm), 30 pWHz^−1/2^ in THz band (*λ* = 300 μm), 3.87 pWHz^−1/2^ in FIR band (λ = 10.3 μm), 4.5 pWHz^−1/2^ in MIR band (*λ* = 4.33 μm), and 8.68 pWHz^−1/2^ in NIR band (*λ* = 870 nm) [25,26,57].

### 4.3. Concetration Approach

Regarding the N-type chemical carrier doping of the CNT film employed in this chapter, the dopant concentration further governs the PTE conversion sensitivities. In terms of PTE conversion, the following equation breaks down the NEP [25]:(7)$${\mathrm{NEP}}^{-1}⟺\frac{S^{2}}{R_{\mathrm{Electrical}}\times k}\times A$$
where *A* is the photo-absorption ratio. This means that employing the dopant concentration that maximizes the term on the right-hand side of Equation (7) yields the smallest NEP value, namely the highest photo-detection sensitivity. In the above model, the photo-induced thermal gradient across the CNT film channel is proportional to the photo-absorption ratio and thermal resistance (inversely proportional to the thermal conductivity). The term on the right-hand side of Equation (7) also corresponds to multiplication of the photo-absorption ratio and the typical dimensionless ZT [25]:(8)$$\mathrm{ZT}=\frac{S^{2}\sigma }{k}T\propto \frac{S^{2}\sigma }{k}\propto \frac{S^{2}}{R_{\mathrm{Electrical}}k}$$

Based on the above considerations, Figure 4h–k shows the changes in each material property: thermal conductivity (Figure 4h), electrical resistance (Figure 4i), photo-absorption ratio (Figure 4j,k), and Seebeck coefficient [25] for the respective dopant concentrations. Therefore, Figure 4l,m maps the NEP value transition of the CNT-film-based photo-detector against the dopant concentration, which is in good agreement with the material-property-driven PTE conversion efficiency. The obtained behaviors indicate that the change in the thermal conductivity dominantly affects both the PTE conversion and photo-detection sensitivity. These assessments lead to proper selection of the dopant concentration (0.4 mol/L in this case) [25], and this process maximizes the photo-detection sensitivity of the CNT-film-based PTE sensor.

To further enhance the photo-sensitivity of the PTE sensor, P-type carrier doping of the semiconducting CNT film may play a crucial role in the device design strategies. The doping of semiconducting CNT films suppresses their inherent higher electrical resistance and enhances their photo-absorption characteristics [58,59]. Therefore, utilizing doped semiconducting CNT films would efficiently expand high-efficiency thermoelectric conversion to photo-detection with suppressed thermal noise.

## 5. Applications: Ubiquitous Non-Destructive Inspections

Based on the PTE design strategies, the CNT-film-based devices advantageously satisfy sensitive broadband photo-detection. As introduced in Section 1, sensitive broadband photo-detector operations facilitate monitoring of objects consisting of composite and layered materials by aggregating multi-spectral optical information specific to each component. This chapter demonstrates ubiquitous non-destructive inspections that function regardless of the operating environment: the structures and locations of target objects [25,26].

### 5.1. Mechanical Durability and Optical Stability

Among the aforementioned PTE device designs of the CNT-film-based photo-detectors, employing the PN-junction structure leads to ubiquitous non-destructive inspections. In particular, developing shape-deformable and shape-conformable device configurations allows omni-directional and blind-spot-free 3D monitoring by surrounding target objects with image sensor sheets without bulky rotation stages [25,26]. To this end, this section evaluates the mechanical durability and associated optical stability of the CNT-film-based PTE sensor.

#### 5.1.1. Stretchability

Figure 5a–e shows the development of the CNT-film-based PTE sensor sheet. The sheet integrates the PN-junction-structured flexible CNT thin-film channel with 8 µm thick polyurethane supporting or sealing substrates and rubber-like readout electrodes (Figure 5a,b) [26]. The electrode consists of a mixture paste of silver nanoparticles and binder resins. In this device structure, a partially patterned epoxy resin on the substrate (Figure 5c) serves as a rigid stiffener. Resin is deposited on the rear side of the CNT film across the substrate (Figure 5d), and FEM simulation clarifies a local distortion-free condition of the stiffened channel (Figure 5e) [26]. Such partial rigid-soft patterning realizes mechanical durability and optical stability of the device against strain. As shown in Figure 5f,g, the existence of the stiffener contributes to more durable device stretching compared with an unstiffened one by avoiding typical disconnections at channel–electrode boundaries [26]. In addition to this mechanical durability, the partial rigid-soft patterning is essential in avoiding significant strain-induced degradation of the device’s thermal noise (Figure 5h). Here, the electrical resistance values of the CNT film channel and electrode are approximately 1 kΩ and sub-1 Ω, respectively, and that of the channel dominantly accounts for the total electrical resistance of the device [26]. These strategies result in shape-deformable and shape-conformable photo-detection in which the CNT-film-based PTE sensor sheet suppresses its sensitivity degradation by at most 20% increments of NEP values under 70–280% multiaxial repetitive strain (Figure 5i,j) [26]. The CNT-film-based PTE sensor sheet then collectively satisfies the sensitive broadband photo-detection and freely adaptable mechanical deformations [60,61]. While the existing flexible or stretchable photo-detectors mostly function in narrowband operations with lower sensitivities compared with the solid-state types [62,63,64,65], the CNT-film-based PTE sensor sheet plays an advantageous role.

#### 5.1.2. Flexibility

Using the above stretchable device design, the CNT-film-based PTE sensor sheet simultaneously exhibits sufficient mechanical durability and the associated optical stability against folding, namely flexibility. Figure 5k shows the photo-detection with the device in each bending radius from the flat state to the folded state. The figure indicates that the CNT-film-based PTE sensor sheet consistently and stably maintains the minimum NEP value of approximately 5 pWHz^−1/2^ during device folding [24,25]. As mentioned previously, such device flexibility and stretchability are essential for imaging of 3D objects at omni-directional viewing angles. In addition, these device features further contribute to monitoring of transformable objects by firmly wrapping them with a CNT-film-based PTE sensor sheet, which stably follows dynamic deformation behaviors modulated by the inherent softness of the targets.

#### 5.1.3. Thermal Property

In addition to the mechanical properties, Figure 5l shows the thermal durability and the associated optical stability of the device. The CNT-film-based PTE sensor sheet consistently and stably maintains the minimum NEP value of approximately 5 pWHz^−1/2^ during device annealing, even at approximately 200 °C [24,26]. Although the reference temperature gradually increases during this evaluation, photo-detection under the operating mechanism of the PTE effect generates additional local temperature increments at the PN junction associated with photo-absorption [26]. This means that the temperature in the photo-detection interface is the highest throughout the CNT film during photo-detection, and the photo-induced thermal gradient across the channel and the associated PTE responses are uniform regardless of the operating temperature. The obtained results indicate that the device itself and the PN junction formed via chemical liquid coating are still functional at higher temperatures. These features suggest the suitability of the CNT-film-based PTE sensor sheet for use as an imaging inspection device, even outdoors, with year-round climate changes. In addition, the above device properties ensure that a variety of fabrication processes are available for designing CNT-film-based PTE sensor sheets, for example, thermal lamination.

### 5.2. Active Imaging

The device characteristics introduced in the previous section indicate that employing the CNT-film-based PTE sensor sheet is widely and ubiquitously adaptable for use in non-destructive monitoring. This section demonstrates multi-functional non-destructive monitoring specific to the broadband multi-wavelength evaluation of composite layered objects based on the photo-absorption of the CNT film and the unmanned omni-directional testing of 3D targets even in a difficult-to-access location owing to the freely attachable and deformable configurations of the sensor sheet [25].

#### 5.2.1. Mechanism

Use of the CNT-film-based PTE sensor sheet allows flexible switching of the reflective or transmission monitoring mode [25]. As structures of 3D objects are mostly solid- or hollow-types, reflective-mode eases monitoring of the former, and the transmission mode handles the latter. Therefore, developing both reflective and transmission types of monitoring modules is essential for non-destructive inspection of 3D objects. For the CNT-film-based PTE sensor sheet, combining 3D printing techniques plays an important role in designing flexibly mode-switchable monitoring modules.

Figure 6a,b develops the respective reflective- and transmission-mode monitoring modules with the CNT-film-based PTE sensor sheet and describes their operating mechanisms. The modules integrate 3D-printed supporting resin bodies and the CNT-film-based PTE sensor sheet. The reflective-module first employs a hemicylinder supporting resin body and firmly attaches the CNT-film-based PTE sensor sheet to the inside of an arc: reflective multi-view stereoscopic capsule PTE imager (Figure 6a) [25]. The module has a window in the resin body to insert external photo-sources. This module scans columnar objects inside the arc during external irradiation. The CNT-film-based PTE sensor sheet consists of multiple-pixels, and each of them comprehensively detects reflection signals against external irradiation to the inspection objects. On the other hand, the transmission module employs a columnar resin supporter and firmly wraps the CNT-film-based PTE sensor sheet around the body: transmissive multi-view stereoscopic PTE endoscope (Figure 6b) [25]. This module scans inside cylindrical objects, and each pixel comprehensively detects transmission signals through the targets against external large-area irradiation.

#### 5.2.2. Reflective System

Figure 6c–e shows examples of the broadband multi-wavelength and multi-view non-destructive monitoring of a composite and layered 3D object [25]. This demonstration employs the reflective multi-view stereoscopic capsule PTE imager and the target consists of a metallic inner column and a plastic opaque outer shell. The inner column conceals light-absorbing impurities on its surface, and there are oil droplets on the outer shell (Figure 6c). In these situations, the module non-destructively extracts hierarchical 3D images of each layer via reflective multi-view stereoscopic capsule imaging in sub-THz and NIR bands. The NIR irradiation first visualizes both faint reflection signals from the shell and their local attenuation due to absorption with oil. On the other hand, the higher transmissive sub-THz irradiation captures the inside of the target through the outer shell and attached oil. The obtained sub-THz image then visualizes both reflection signals from the metallic column and their local attenuation due to absorption with impurities (Figure 6d). Based on these differences in optical properties of the target’s respective components in sub-THz and NIR bands, images in each wavelength extract characteristics of each layer. By incorporating the above images, the module simply reconstructs multi-view 3D structures of the target object (Figure 6e) [25].

The concept demonstrated here potentially handles universally structured objects under proper selection of the imaging wavelength freely in ultrabroadband ranges with the optical properties of the module. Furthermore, integrating miniature photo-sources, such as short-wavelength-IR LEDs, long-wavelength-IR quantum cascade lasers [66], THz resonant tunneling diodes [67], and sub-THz Gunn diode chips [68], with the reflective module by embedding them into the body’s window potentially allows compact and mobile systemizations for ubiquitous inspections. The potential inspection and reconstruction targets in the industrial field are multilayered columnar components, such as power transmission lines or communication cables, which require high safety standards.

#### 5.2.3. Transmission System

Figure 6f,g demonstrates robot-assisted self-driving multi-view stereoscopic 3D PTE endoscopy in an unmanned and non-destructive manner [25]. The CNT-film-based PTE sensor sheet is freely attachable, and wrapping the sheet around the body of a self-driving unit allows robot-assisted unmanned monitoring operations. The inspection object is an L-shaped and confined miniature tunnel model, and the self-driving endoscope comprehensively scans the inside of the target during external irradiation (Figure 6g). The tunnel conceals tiny breakages randomly on its outer wall and the self-driving endoscope comprehensively and omni-directionally detects local transmission signals against external irradiation through the defects. Together with these transmission signals, considering the spatial coordinates of each pixel of the endoscope based on information of driving positions and outer shape of the unit results in non-destructive unmanned 3D reconstruction of the defective tunnel (Figure 6g) [25].

Here, the endoscope collectively satisfies broadband multi-view monitoring and potentially enriches the existing field of endoscopic technology [69,70,71] for single-wavelength band imaging (mainly in Vis). The presented approach also assesses the quality of gas and water pipes under non-destructive multi-view operations [25,72]. The concept demonstrated here also provides advantageous high-maneuverable operations of the multi-view stereoscopic PTE monitoring module, potentially realizing easier and safer access to targets over manned testing. This means that further functionalization of the monitoring system is feasible by combining the module with cutting-edge robot-assisted operating methods, such as multi-axis movement [73], wall climbing [74], and underwater or aerial swimming [75,76]. These testing demonstrations against industrial components or infrastructure models would contribute to future mitigation of secondary damage after natural disasters (e.g., power outages (underground gas pipes, suspended power transmission lines) or water cuts (underground water pipes), traffic blockage (aerial or sea bridges), and communication failure (undersea cables)).

### 5.3. Passive Imaging

In addition to the above multi-functional non-destructive imaging inspections in the active style, passive sensing (external photo-source-free) configurations significantly contribute to development of ubiquitous monitoring platforms in different ways. Therefore, this section introduces use of the CNT-film-based PTE sensor sheet in chemical evaluations as an example of a passive-style application [26].

Chemical monitoring plays a crucial role in anticipating possible upcoming natural or social events by continuously communicating diverse environmental information and its changes [77,78]. Examples of such concepts are as follows: quantitative evaluation of specific chemicals or identification of contaminants in industrial equipment [79,80], wastewater treatment [81], and food or beverage products [82,83]. Focusing on methodologies, typical chemical monitoring approaches include centrifugation measurements [84], reagent reaction observation [85], biomarker tests [86], and biofluid secretion monitoring [87]. However, these methods assume sampling or labeling processes and examine the chemicals within the testing packages rather than handling the targets in their natural behaviors. Such invasive and static operations potentially hinder target-friendly and on-site chemical evaluation. As mentioned in the previous section, even poorly accessible infrastructure or facilities continuously have certain impacts on our daily lives and social systems. Therefore, on-site operation is indispensable. To this end, non-sampling or label-free chemical monitoring is feasible via photo-sensing measurements (typically in Vis–UV bands) [88,89]. Nonetheless, the low transparency of such bands to opaque containers, for example, plastic or silicon, hinders non-destructive evaluations.

Therefore, photo-irradiation in higher-transparency THz–IR bands [90,91] relaxes the above technical difficulties in realizing non-destructive and on-site chemical evaluations. Indeed, THz sensing and imaging with the CNT-film-based PTE sensor sheet of solid or gas samples simply analyze the material properties of the targets in a non-destructive, non-sampling, and label-free manner [26,92,93]. In addition to solid or gas samples, handling liquid targets is essential to develop a ubiquitous non-destructive monitoring platform because chemicals often dissolve in water, oil, alcohol, etc. As liquid samples generally absorb THz–IR irradiation well and cause signal shortages for fundamental reflective or transmissive measurements [94], natural emission in these bands from the target itself, namely blackbody radiation (BBR) based on Planck’s law, is noteworthy. Such a passive monitoring configuration employing the BBR of liquids not only facilitates non-destructive on-site evaluations of in-liquid chemicals but also functions in compact experimental systems. This means that the effective use of the target’s BBR allows fundamental photo-sensing or imaging measurements without employing external sources and associated optical systems.

Based on these discussions, Figure 7a,b describes the operating mechanism of the BBR-based non-destructive on-site passive monitoring of in-liquid chemicals with the CNT-film-based PTE sensor sheet. The experimental system only requires wrapping the CNT-film-based PTE sensor sheet firmly around a soft liquid flowing or circulating tube (Figure 7a) [26]. Furthermore, the shape-conformable mechanical durability and associated optical stability of the sheet dynamically monitor the flowing targets in transformable tubes well by following the container deformations associated with inherent liquid fluidity. Therefore, the presented technique potentially provides anyone, not just experts, with opportunities to sustainably monitor in-liquid chemicals flowing in bellows hoses, medical rubber tubes, and plants in a target-friendly manner. The CNT-film-based PTE sensor sheet continuously detects the BBR of the solvent. Here, dissolution of the chemical substance induces reductions in the PTE responses of the device owing to local absorption of the solvent’s BBR with in-liquid chemicals (Figure 7b) [26]. This means that comparison of the reduction magnitudes of the device’s PTE response induced by BBR from the solvent before and after dissolving the chemicals leads to simple solution concentration monitoring (Figure 7c). For example, Figure 7d demonstrates the detectivity of the presented approach against a glucose solution, which currently ranges from 50 to 20,000 mg/dL [26]. The obtained performance potentially satisfies the sensitivity requirement for diverse chemical monitoring applications, including diabetes diagnosis [95] and plant or fruit sugar content measurements [96]. Meanwhile, developing a 360° around-view PTE imaging module (Figure 7e) captures and reconstructs the concealed liquid circulation in a non-destructive manner (Figure 7f–i: inspired by the previous section). By incorporating the above findings and techniques, the presented BBR-based non-destructive on-site passive monitoring of in-liquid chemicals omni-directionally and dynamically visualizes changes in the concentration of the glucose solution (Figure 7j) [26]. This system can handle arbitrary liquid-soluble chemical substances in principle. As the current demonstrations focus on single-component solutions, possible use cases include quality monitoring and impurity identification of manufacturing sites of beverages (e.g., sugar water), industrial chemicals (e.g., chlorine bleach), and medical practice fields (e.g., saline infusion).

## 6. Conclusions

This paper summarized the fundamentals and applications of CNT-film-based PTE sensors. The material properties and structural nature of CNT films play an essential role in both fundamental device design strategies and multi-functional non-destructive inspection applications. Each presented factor and their combinations bridge the gap between the investigation fields of nanocarbon materials, thermoelectrics, PTE conversion, flexible and stretchable electronics, photo-sensing or imaging, and non-destructive inspections. The above series of mechanism modeling and demonstrations are examples of the synergistic concept between photo-absorption and thermoelectric conversion. This means that further combinations with closely related investigation fields maximize the potential of PTE conversion with CNT film: ink-jet-based all-printable device fabrication, integration with mechanically stretchable and optically tunable broadband filter sheets, computer-vision-driven image processing, etc.

## Figures and Tables

**Figure 1 micromachines-14-00061-f001:**
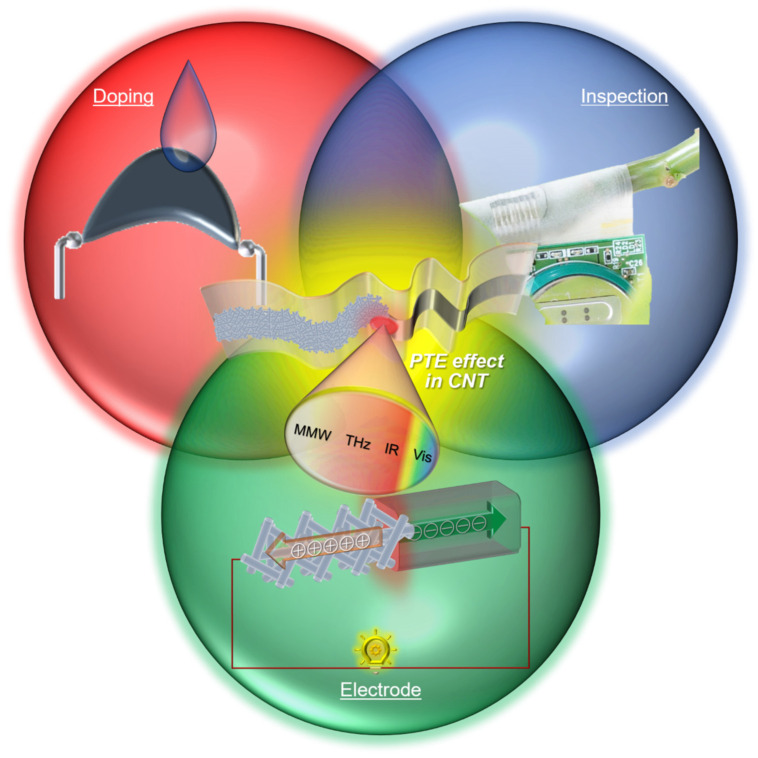
Conceptual diagram of developing the CNT-film-based PTE sensors. Reproduced with permission from [24] Copyright 2021, the authors, published by Wiley-VCH. Reproduced with permission from [25] Copyright 2021, the authors, published by Springer Nature. Reproduced with permission from [26] Copyright 2022, the authors, published by American Association for the Advancement of Science.

**Figure 2 micromachines-14-00061-f002:**
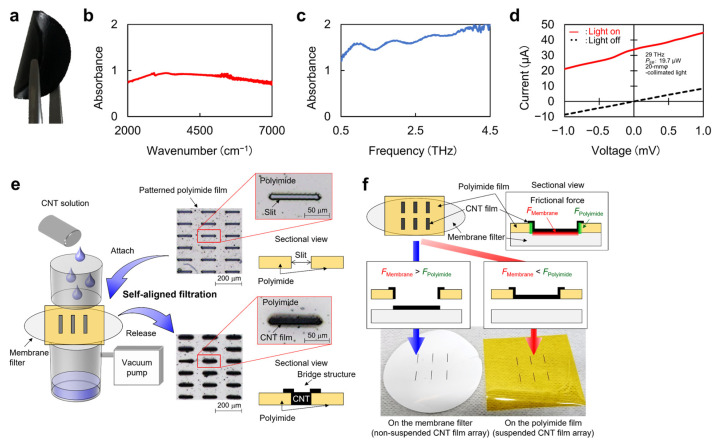
Fundamental properties of CNT films as the device material for the PTE sensors. (**a**) Photograph of a CNT film. (**b**) IR absorption characteristic of the CNT film. (**c**) THz absorption characteristic of the CNT film. (**d**) Current-voltage characteristics with and without light irradiation to the CNT film PTE sensor. (**e**) Diagram of the self-aligned CNT filtration process. (**f**) Examples of selective CNT film patterning. (**a**,**d**): Reproduced with permission from [24] Copyright 2021, the authors, published by Wiley-VCH. (**b**): Reproduced with permission from [26] Copyright 2022, the authors, published by American Association for the Advancement of Science. (**c**): Reproduced with permission from [25] Copyright 2021, the authors, published by Springer Nature. (**e**,**f**): Reproduced with permission from [36] Copyright 2021, the authors, published by Wiley-VCH.

**Figure 3 micromachines-14-00061-f003:**
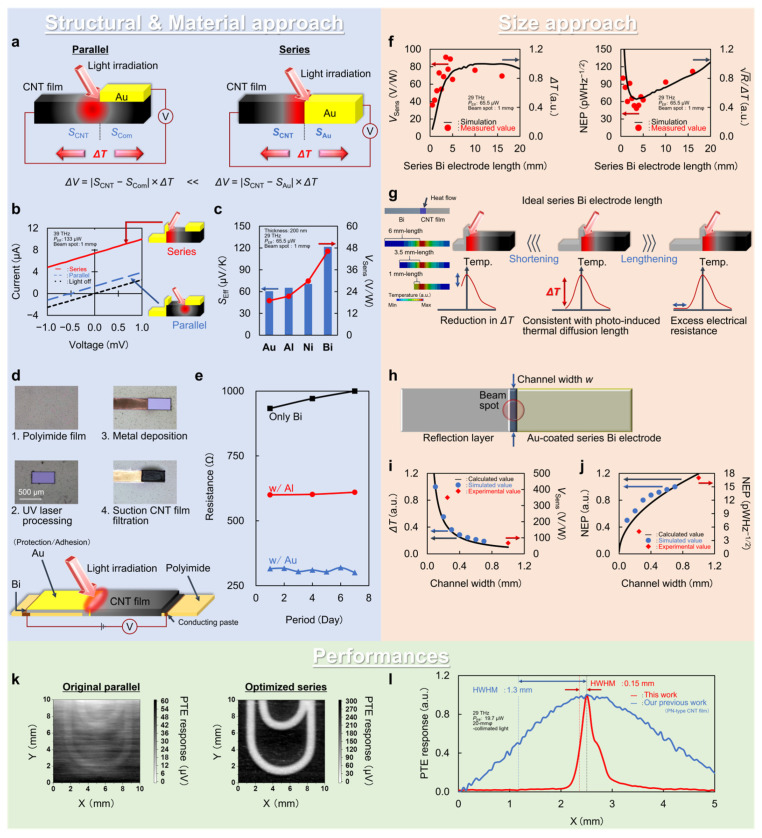
PTE effect in the CNT film channel–metal electrode boundary. (**a**) Operating mechanism. (**b**) Change in the device’s photo-current responses depends on the electrode structure. (**c**) Changes in the device’s effective Seebeck coefficients and PTE responses for different metal electrode materials. (**d**) Photographs of the device fabrication process. (**e**) Time course of changes in the device’s electrical resistances for different metal electrode lamination. (**f**) Changes in the device’s photo-detection sensitivities against the electrode length. (**g**) Schematic models for optimizing the device’s electrode length. (**h**) Schematic model for optimizing the device width. (**i**,**j**) Changes in the photo-detection sensitivities against the device width. (**k**) Comparison of the PTE images. (**l**) Comparison of the PTE response distributions across the device area. (**a**–**l**): Reproduced with permission from [24] Copyright 2021, the authors, published by Wiley-VCH.

**Figure 4 micromachines-14-00061-f004:**
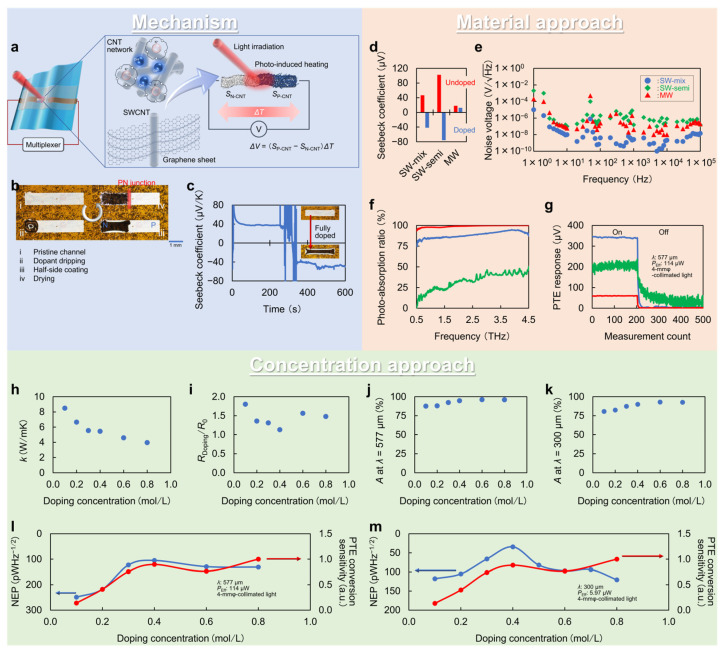
PTE effect in the CNT film channel’s PN-junction. (**a**) Operating mechanism. (**b**) Process of the N-type chemical carrier doping. (**c**) Change in the Seebeck coefficient of the CNT film before and after doping. (**d**) Changes in the Seebeck coefficients, (**e**) the noise voltage, and (**f**) the photo-absorption ratio in sub-THz frequency region of the device channels for different CNTs types. (**g**) Comparison of the PTE responses of each type of the CNT-film-based photo-detector. (**h**) Changes in the thermal conductivity, (**i**) electrical resistance, and photo-absorption ratios in (**j**) sub-THz frequency region and (**k**) THz frequency region. (**l**) Changes in the NEP values and PTE conversion sensitivities in sub-THz frequency region and (**m**) THz frequency region. (**a**–**m**): Reproduced with permission from [25] Copyright 2021, the authors, published by Springer Nature.

**Figure 5 micromachines-14-00061-f005:**
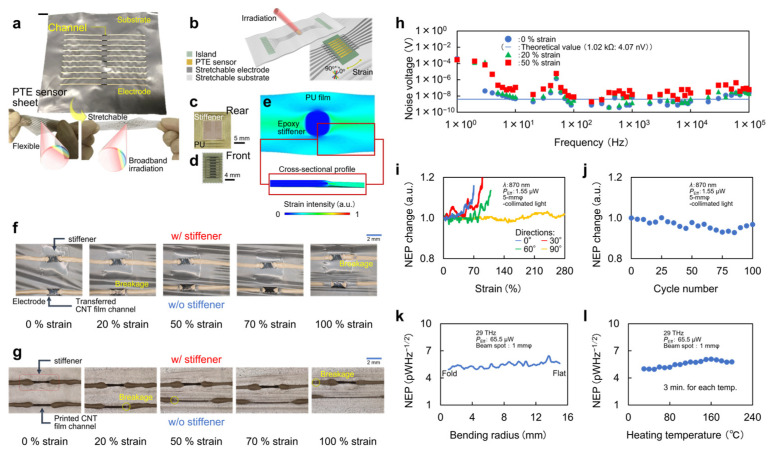
Mechanical and thermal durability and the associated optical stability of the CNT-film-based PTE sensor sheet. (**a**) Photograph and (**b**) schematic of the device. (**c**) Photograph of the rigid-soft patterning. (**d**) Photograph of the CNT film channel located above the partial stiffener. (**e**) FEM simulation of strain distribution on the device. (**f**) Comparisons of the device stretchability with and without the partial rigid stiffener with transferred CNT film channels and (**g**) printed CNT film channels. (**h**) Stability of the device thermal noise against strains. (**i**) Changes in NEP values of the device against multiaxial strains. (**j**) Stability of the device’s NEP values against the repetitive stretching. (**k**) Stability of the device’s NEP values in each bending radius during the sheet folding. (**l**) Thermal stability of the device’s NEP values. (**a**–**j**): Reproduced with permission from [26] Copyright 2022, the authors, published by American Association for the Advancement of Science. (**k**,**l**): Reproduced with permission from [24] Copyright 2021, the authors, published by Wiley-VCH.

**Figure 6 micromachines-14-00061-f006:**
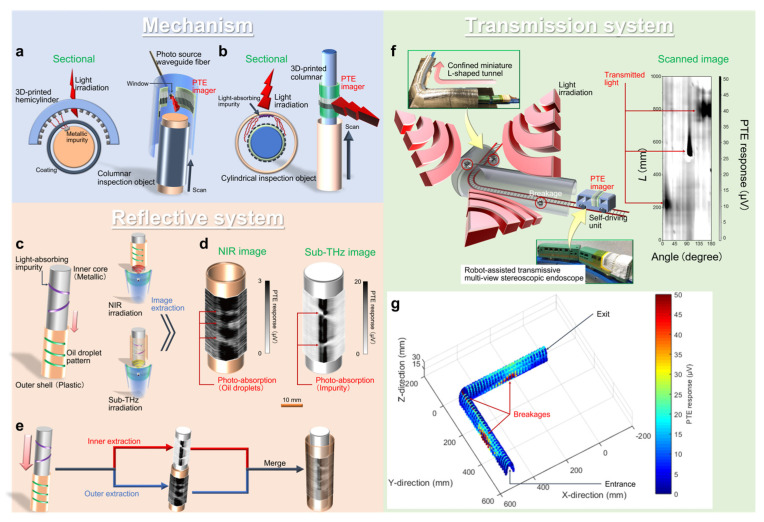
Demonstrations of active-style non-destructive inspections with the CNT-film-based PTE sensor sheet. (**a**) Operating mechanism of the monitoring modules in reflective and (**b**) transmission systems. (**c**) Simple diagram for the multi-wavelength reflective multi-view stereoscopic capsule PTE imaging. (**d**) Extracted reflective multi-view PTE images in NIR and sub-THz bands. (**e**) Simple reconstruction of the 3D multi-layered structure. (**f**) Simple diagram for the robot-assisted self-driving transmission multi-view stereoscopic PTE endoscopy. (**g**) 3D PTE image reconstruction of a defective L-shaped and confined miniature tunnel model. (**a**–**g**): Reproduced with permission from [25] Copyright 2021, the authors, published by Springer Nature.

**Figure 7 micromachines-14-00061-f007:**
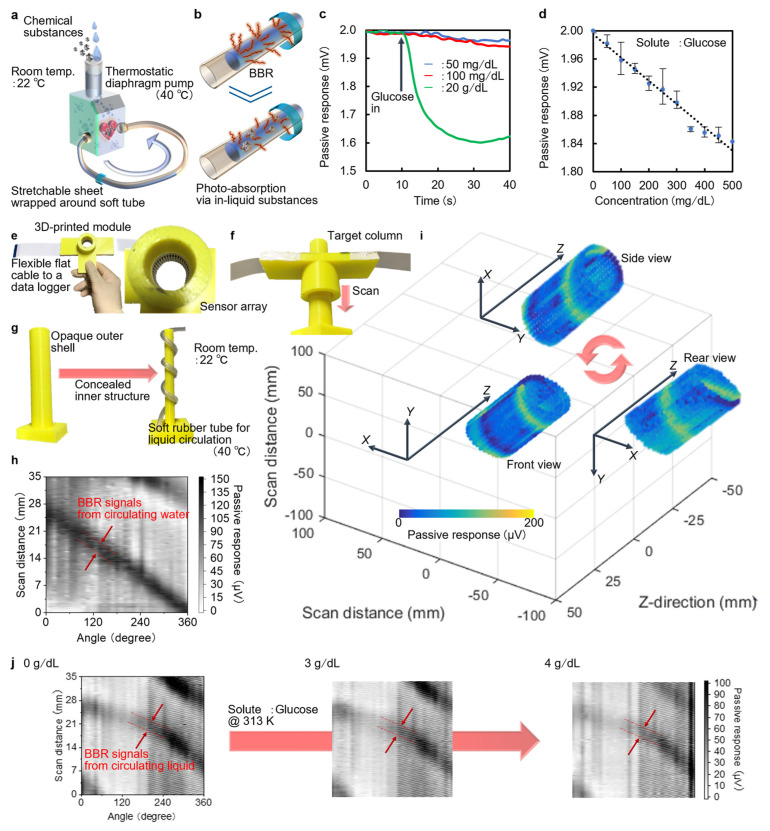
Demonstrations of passive-style non-destructive, on-site, and dynamic monitoring of in-liquid chemicals. (**a**) Schematic of the experimental setup. (**b**) Operating mechanism. (**c**) Changes in the device’s PTE responses against the amount of glucose powder dissolution to the circulating liquid. (**d**) Concentration monitoring of the glucose solution. (**e**) Photographs of the porTable 360° around-view PTE imager with the CNT-film-based PTE sensor sheet. (**f**) Experimental setup for the BBR-based 360° around-view passive imaging of the spirally circulating liquid. (**g**) Photographs of the target. (**h**) 360° around-view PTE image. (**i**) 3D PTE image reconstruction of the target. (**j**) Comparison in BBR-induced 360° around-view PTE images of the spirally circulating glucose solution against concentration. (**a**–**j**): Reproduced with permission from [26] Copyright 2022, the authors, published by American Association for the Advancement of Science.

## Data Availability

Not applicable.

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
