# Peer review of "Recent Progress in Development of Carbon-Nanotube-Based Photo-Thermoelectric Sensors and Their Applications in Ubiquitous Non-Destructive Inspections"

_micromachines, 2022, doi:10.3390/mi14010061_

Round 1
Reviewer 1 Report
Minor Revision:
In the review study, the authors discussed recent progress in the development of carbon nanotubes-based 3 photo-thermoelectric sensors and their applications in ubiquitous non-destructive inspection, which effectively combines photo absorption-induced local heating and the associated thermoelectric conversion for uncooled and broadband photo-detection. The present study provides detailed insight for indorsing the contributions of PTE device design strategies with carbon nanotube (CNT) films to demonstrate multi-functional non-destructive inspections, and the theoretical background study clarify the respective contributions from material, structural, mechanical, and optical approaches to multifunctional performances of the CNT film PTE sensors. The manuscript is well organized. At present, I feel that this manuscript is suitable for publication in this journal. However, I suggest some minor modifications before publication.
1. For introduction part, it is suggested to include/mention more discussion based on main objective of present study.
2. Line 75-79, It is suggested to add some more explanation about photo-thermoelectric effect in Introduction part.
3. Line 146, reference should be about PTE effect in channel-electrode junctions.
4. Line 276, The description about PTE in PN junction should be properly described by reference. Similarly, the whole part of “Application: Ubiquitous non-destructive inspection” should mention with proper references.
5. All the equations used in the manuscript should be mentioned with accurate references.
Reviewer 2 Report
This is a quite useful review on the topic of the fundamentals and applications of CNT film-based PTE sensors. I consider it is well written and researched, the references are overall recent and relevant and the figures are interesting and informative. I recommend it to be published in its current form.
